# COVID-19 Vaccination Reduces Lower Limb Amputation Rates and Mortality Rate in Patients with Pre-Existing Peripheral Vascular Disease Based on TriNetX Database

**DOI:** 10.3390/vaccines13090969

**Published:** 2025-09-12

**Authors:** Shiuan-Tzuen Su, Yu-Hsuan Huang, Jing-Yang Huang, James C.-C. Wei

**Affiliations:** 1Department of Surgery, Chung Shan Medical University Hospital, No. 110, Sec. 1, Jianguo N. Rd., South District, Taichung 40201, Taiwan; jasonsquare1000@gmail.com; 2School of Medicine, Chung Shan Medical University, Taichung 40201, Taiwan; 3Department of Ophthalmology, Asia University Hospital, Taichung 40201, Taiwan; jasonsquare3000@gmail.com; 4Department of Optometry, Asia University, Taichung 40201, Taiwan; 5School of Medicine, College of Medicine, China Medical University, Taichung 40201, Taiwan; 6Center for Health Data Science, Chung Shan Medical University Hospital, Taichung 40201, Taiwan; wchinyang@gmail.com; 7Institute of Medicine, Chung Shan Medical University, Taichung 40201, Taiwan; 8Division of Allergy, Immunology and Rheumatology, Department of Internal Medicine, Chung Shan Medical University Hospital, Taichung 40201, Taiwan; 9Graduate Institute of Integrated Medicine, China Medical University, Taichung 40201, Taiwan; 10Office of Research and Development, Asia University, Taichung 40201, Taiwan

**Keywords:** COVID-19, lower limb, amputation, mortality, SARS-CoV-2

## Abstract

**Background:** Unvaccinated individuals with peripheral arterial occlusive disease (PAOD) are more likely to develop acute limb ischemia (ALI) following severe acute respiratory syndrome coronavirus 2 (SARS-CoV-2) infection. We assessed the protective effect of the COVID-19 vaccine in preventing ALI in PAOD patients with SARS-CoV-2 infection. **Methods:** This retrospective cohort study was conducted using the United States TriNetX (Cambridge, MA, USA), using patients with PAOD who were diagnosed with SARS-CoV-2 infection between 1 November 2020 and 31 December 2023. Propensity score matching was performed to adjust for demographic variables, lifestyle factors, medical utilization, and comorbidities. Cox proportional hazards models were used to compare the two matched cohorts. Kaplan–Meier analysis estimated the 3-year cumulative probability of lower limb amputation incidence. We selected 12,948 PAOD patients who received the COVID-19 vaccine and 44,064 PAOD patients who were unvaccinated against COVID-19. **Results:** A total of 11,822 pairs of COVID-19 vaccinated PAOD patients and unvaccinated individuals were compared. The mean (SD) age was 66.5 (14.1) years; there were 4849 male patients (41%) and 6569 female (55.6%) compared to unvaccinated PAOD patients, and those who received the COVID-19 vaccine had a significantly lower risk of 3-year all-cause mortality (log-rank test, *p* < 0.001; hazard ratio (HR) was 0.857; 95% CI, 0.796–0.922) and lower limb amputation (log-rank test, *p* = 0.001, HR = 0.716; 95% CI, 0.587–0.873), though there was no significant difference in ischemic stroke (log-rank test, *p* = 0.174; HR = 0.958; 95% CI, 0.902–1.019). **Conclusions:** This study found that patients who received the COVID-19 vaccine had a significantly lower risk of 3-year all-cause mortality and lower limb amputation, though there was no significant difference in ischemic stroke.

## 1. Introduction

Peripheral arterial occlusive disease (PAOD) is characterized by atherosclerosis of the peripheral blood vessels. Major risk factors include hypertension, diabetes, dyslipidemia, advanced age, and smoking [1]. The prevalence of PAOD ranges from 3% to 10% and increases with advancing age [1]. Most cases of PAOD may progress to coronary artery disease (CAD) [1,2], leading to cardiovascular death.

The majority of PAOD cases are asymptomatic, while symptomatic PAOD can be categorized into intermittent claudication, ischemic pain at rest, or necrosis involving small or large areas [3]. Symptomatic PAOD typically results from acute thrombotic obstruction of blood flow in the lower limbs, leading to pain or tissue necrosis through mechanisms similar to those observed in CAD. Tissue necrosis and infection of the lower limbs require amputation to avoid death from sepsis.

The first infection was reported in Wuhan, China, leading to the severe acute respiratory syndrome coronavirus 2 (SARS-CoV-2) [4]. COVID-19 was not only a type of viral pneumonia but also presented numerous extrapulmonary complications, such as thrombosis, as demonstrated in various studies [5].

Bellosta et al. first reported peripheral arterial occlusion following SARS-CoV-2 infection [6]. In a 2021 single-center study in France, arterial thrombosis occurred in 30 of 531 hospitalized patients [7], and a four-hospital study in New York City identified arterial thrombosis in 365 of 3334 patients [8]. SARS-CoV-2 infection has been linked to acute limb ischemia (ALI) [9], although the underlying mechanism of thrombosis formation remains unclear. Proposed pathways include an inflammatory cascade in which IL-1 upregulation stimulates IL-6, promoting leukocyte infiltration, endothelial injury, and increased fibrinogen production [10], thereby creating a hypercoagulable and prothrombotic state [6,8,10]. Additionally, the virus could infect endothelial cells through angiotensin-converting enzyme 2 (ACE2) receptors, with inflammation causing endothelial injury, which might also subsequently contribute to the formation of free-floating aortic thrombus [11,12], which can precipitate ALI.

ALI following SARS-CoV-2 infection showed a male predominance, accounting for approximately 50–90% of reported cases [6,7,8,9,13,14,15,16,17]. Previous studies have found high mortality associated with post-infection ALI, ranging from 23 to 40% [6,7,8,9,13,14,15,16,17]. Two studies specifically noted high amputation rates [13,14]. In patients with pre-existing PAOD, the occurrence of ALI might further increase mortality. Recent work by Xie et al. indicated that unvaccinated individuals had a higher likelihood of developing ALI following SARS-CoV-2 infection [5], although the study was limited by a small sample size (36 individuals). The protective effect of COVID-19 vaccination against ALI among PAOD patients who acquire a SARS-CoV-2 infection remains unknown. To evaluate the potential benefit of the COVID-19 vaccine in improving prognosis in PAOD patients, we conducted a study by using the United States (US) Collaborative Network from 48 healthcare organizations (HCOs) within the TriNetX Research Network. We hypothesized that prior COVID-19 vaccination may provide protection for PAOD patients.

## 2. Methods

### 2.1. Data Source

The dataset utilized in this study was collected on 17 July 2024 from the TriNetX US Network. TriNetX is a global federated health research platform providing access to electronic medical records (including diagnoses, procedures, medications, laboratory values, and genomic information) from large HCOs. This analysis used the US Collaborative Network, which included 66 HCOs at the time of query. TriNetX, LLC complies with the Health Insurance Portability and Accountability Act (HIPAA) and other applicable data privacy regulations of contributing HCOs. The platform is ISO 27001:2013 certified and maintains an Information Security Management System (ISMS) to support HIPAA Security Rule compliance and safeguard healthcare data.

All data displayed on the TriNetX Platform in aggregate form or included in any patient-level dataset generated by the platform are de-identified in accordance with the de-identification standards outlined in Section §164.514 (a) of the HIPAA Privacy Rule. De-identification process is verified through a formal determination by a qualified expert as defined in Section §164.514 (b) [1] of the HIPAA Privacy Rule. Consequently, this study involved only de-identified patient records and did not involve the collection, use, or transmission of individually identifiable information. Additionally, the use of TriNetX for this research was approved by the Institutional Review Board of Chung Shan Medical University Hospital (CSMUH), Taichung, Taiwan, under the approval number CS2-21176.

### 2.2. Study Population

Initially, we identified 459,851 adult patients aged ≥18 years who had ≥2 medical encounters for peripheral vascular disease (ICD-10: I73.0, I73.1, I73.8, and I73.9) or arterial embolism and thrombosis diseases (ICD-10: I74) between 1 January 2015 and 31 December 2019. From this population, we selected 57,012 SARS-CoV-2 infection cases (defined as ICD-10-CM U07.1 or PCR positive for SARS-CoV-2 RNA) that occurred between November 2020 and December 2023. Since the COVID-19 vaccine was first approved in the US around November 2020, we only included patients who were infected with SARS-CoV-2 during the period from November 2020 to December 2023. The index date was the first documented SARS-CoV-2 infection.

### 2.3. COVID-19 Vaccine Group

We identified 12,948 PAOD patients who received at least one COVID-19 vaccination before the index date. After excluding individuals with any pre-index records of lower-limb amputation (ICD-10 diagnoses or procedure codes), we included 11,956 vaccinated patients in the analysis. For sensitivity analyses, we classified vaccination by dose count, 0 doses (unvaccinated), 1 dose, 2 doses, and ≥3 doses (booster). The vaccine type was also identified using product codes (CPT/CVX/RxNorm): BNT (BNT162b2; Pfizer–BioNTech), Moderna (mRNA-1273), and adenoviral vector (e.g., Ad26.COV2.S).

### 2.4. Unvaccinated Group

A total of 44,064 PAOD patients had no record of COVID-19 vaccination before the index date. After excluding those with any pre-index documentation of lower limb amputation (including ICD-10 diagnosis and procedure codes), 40,324 unvaccinated patients were included in the analysis.

### 2.5. Outcomes

The primary outcome was the 3-year risk of lower limb amputation, comparing patients with PAOD who were COVID-19 vaccinated before SARS-CoV-2 infection with those without prior vaccination. The secondary outcome was the 3-year risk of ischemic stroke. For each participant, follow-up began on the index date (defined as the date of COVID-19 diagnosis) and ended at the earliest of the outcomes event or censoring at the last recorded observation within 3 years of the index date.

### 2.6. Additional Variables

To adjust for potential confounding, our study included covariates measured on the index date or within 6 months prior to the index date. Demographic details comprised age, sex, ethnicity, race, and socioeconomic status. Lifestyle factors included BMI, nicotine dependence as an indicator of smoking habits, and alcohol-related disorders as an indicator of alcohol consumption habits. Baseline medical utilization was captured via preventive medicine services, inpatient encounters, and emergency visits. Additionally, we considered comorbidities including hypertensive diseases, ischemic heart diseases, heart failure, cerebrovascular diseases, diabetes mellitus, dyslipoproteinemia, chronic lower respiratory diseases, liver diseases, chronic kidney disease, chronic ulcers of the lower limb, gangrene, osteoporosis, and rheumatoid arthritis. The definitions of the study covariates are listed in Table 1.

### 2.7. Statistical Analysis

Patient cohorts were propensity score matched on baseline covariates, including demographics, lifestyle factors, medical utilization, and comorbidities. Propensity score matching was performed, and the system generated a propensity score for each patient in each cohort. The propensity score ranges between 0 and 1 and indicates the predicted probability of a patient being in the vaccinated cohort given the patient’s covariates. Logistic regression was used to generate the propensity scores through an implementation of the well-tested standard software package scikit-learn(version 0.24.2 (Python 3.8 environment)). Data were pooled across all HCOs, so all patients in the analysis are represented as rows in one of these matrices. All missing values in each matrix were imputed as the mean from that column, depending on the variable type (binary, categorical, or continuous). We used “greedy nearest neighbor matching” with a caliper of 0.1 pooled standard deviations. A caliper of 0.1 pooled standard deviations of the propensity scores ensures that patients with very different propensity scores are not matched. TriNetX pulls data from a federated data network composed of many healthcare organizations worldwide. Each site computes a covariate matrix for the patients they contribute to the analysis and sends it to a central processing point to be pooled and analyzed as a single matrix. The order of the rows in the matrix should not impact the propensity scores generated for each patient. To eliminate any potential bias, we randomized the order of the records in the covariate matrix.

After propensity score matching, there were 11,822 pairs of PAOD patients with and without the COVID-19 vaccine (Figure 1). The Kaplan–Meier analysis estimated the cumulative probability of lower limb amputation incidence at daily time intervals (Figure 2). To account for patients who exited the cohort during the analysis period and therefore should not be included in the analysis, censoring was applied. In this analysis, patients were censored at the last recorded fact within the time window in the patient’s record. Cox’s proportional hazards model was used to compare the two matched cohorts. The proportional hazard assumption was tested using the generalized Schoenfeld approach. The TriNetX Platform calculated the hazard ratios and associated confidence intervals using R’s Survival package v3.2-3. In sensitivity analyses, we estimated associations by vaccine type (BNT vs. unvaccinated; Moderna vs. unvaccinated; adenoviral vector vs. unvaccinated) using the same matching and time-to-event procedures. We also conducted head-to-head comparisons among vaccinated patients (e.g., BNT vs. Moderna). In addition, we compared each dose category (1, 2, and ≥3 doses) against the unvaccinated group. All statistical tests were conducted within the TriNetX Analytics Platform, with significance set at *p* < 0.05 (two-sided).

## 3. Results

After propensity score matching, 11,822 pairs of COVID-19 vaccinated PAOD patients and unvaccinated individuals were compared (Figure 1). Vaccinated patients had significantly lower risk of 3-year all-cause mortality (log-rank test, *p* < 0.001) and lower limb amputation (log-rank test, *p* = 0.001) than unvaccinated patients, though there was no significant difference in ischemic stroke (log-rank test, *p* = 0.174). The corresponding hazard ratios (HRs) were 0.857 (95% CI, 0.796–0.922) for all-cause mortality, 0.716 (95% CI, 0.587–0.873) for lower limb amputation, and 0.958 (95% CI, 0.902–1.019) for ischemic stroke (Table 2).

Figure 2 presents cumulative incidence curves. Compared with unvaccinated counterparts, PAOD patients who received a COVID-19 vaccine showed a lower cumulative incidence of both all-cause mortality and lower limb amputation. No significant association was observed between vaccination and ischemic stroke.

We conducted analyses stratified by vaccine type, comparing BNT, Moderna, and adenoviral vector vaccines with the unvaccinated cohort, as well as head-to-head comparisons between mRNA vaccines (Appendix A). For all-cause mortality, neither BNT (HR = 0.932; 95% CI = 0.836–1.030) nor Moderna (HR = 0.924; 95% CI = 0.790–1.000) showed a significant difference in risk compared to the unvaccinated group. Similarly, the head-to-head comparison between Moderna and BNT revealed no significant difference in mortality risk (HR = 0.994; 95% CI = 0.850–1.160). The adenoviral vector vaccine group also showed no significant difference in mortality risk versus the unvaccinated group (HR = 1.436; 95% CI = 0.771–2.670). For the risk of ischemic stroke, there were no significant differences observed for BNT (HR = 1.105; 95% CI = 0.967–1.260), Moderna (HR = 1.074; 95% CI = 0.873–1.320), or adenoviral vector vaccines (HR = 0.716; 95% CI = 0.288–1.780) when compared to the unvaccinated cohort. The direct comparison between Moderna and BNT also showed no significant difference (HR = 0.976; 95% CI = 0.799–1.190). For the risk of lower limb amputation, no significant associations were found when comparing BNT (HR = 0.783; 95% CI 0.580–1.050), Moderna (HR = 0.822; 95% CI = 0.505–1.330), or adenoviral vector vaccines (HR = 1.617; 95% CI = 0.386–6.770) with the unvaccinated group. The head-to-head comparison of Moderna versus BNT was also not statistically significant (HR = 0.967; 95% CI = 0.588–1.590).

We further performed analyses stratified by the number of vaccine doses (one, two, or booster dose) for BNT and Moderna vaccines (Appendix A). For all-cause mortality, receiving a BNT booster was associated with a significantly lower risk compared to the unvaccinated group (HR = 0.777; 95% CI = 0.666–0.900). No such association was observed for one dose (HR = 1.144; 95% CI = 0.732–1.780) or two doses (HR = 1.126; 95% CI = 0.926–1.300) of the BNT vaccine. For the Moderna vaccine, no significant association with all-cause mortality was found for any dose category when compared to the unvaccinated group. For the risk of ischemic stroke, no significant associations were identified for any dose category of either BNT or Moderna vaccines when compared with the unvaccinated cohort. For the risk of lower limb amputation, receiving one dose of the BNT vaccine was associated with a significantly lower risk compared to being unvaccinated (HR = 0.155; 95% CI = 0.035–0.690). No other dose categories for either BNT or Moderna vaccines showed a significant association with the risk of lower limb amputation. There was no amputation event recorded in the Moderna single-dose group or its matched unvaccinated cohort.

## 4. Discussion

In a matched cohort of 11,822 COVID-19 vaccinated recipients and 11,822 unvaccinated controls, vaccination was associated with a lower risk of lower limb amputation (HR = 0.716; 95% CI = 0.587–0.873). Additionally, vaccinated individuals demonstrated a lower risk of all-cause mortality (HR = 0.857; 95% CI = 0.796–0.922). Interestingly, the analysis showed no significant relationship between COVID-19 vaccination and the risk of ischemic stroke in this patient population (HR = 0.958; 95% CI = 0.902–1.019). These findings highlight potential protective benefits of COVID-19 vaccination against severe outcomes for patients with PAOD.

SARS-CoV-2 enters host cells via binding of its spike proteins to the ACE2 receptor and proteolytic priming by the host protease TMPRSS2. Following entry into respiratory epithelial cells, SARS-CoV-2 activates the immune system, inducing a broad cytokine response while eliciting a comparatively weak interferon response. The immune activation stimulates proinflammatory Th1 cells and CD14+CD16+ monocytes, amplifying signaling pathways that recruit macrophages and neutrophils to the sites of infection. The resulting feed-forward cascade culminates in a cytokine storm, contributing to severe lung inflammation [18]. SARS-CoV-2 rapidly activates pathogenic Th1 cells to produce granulocyte macrophage colony-stimulating factor (GM-CSF) and interleukin-6 (IL-6). GM-CSF further stimulates CD14+CD16+ monocytes to release tumor necrosis factor-α (TNF-α) and IL-6. Membrane-bound immune receptors, including Fc and Toll-like receptors, contribute to this inflammation amplification, while weak interferon-γ (IFN-γ) responses permit continued cytokine production. Neutrophil extracellular traps also promote cytokine release. The cytokine storm, characterized by elevated IL-6 and TNF-α levels, is linked to dysregulated angiotensin 2 (AngII) signaling. SARS-CoV-2 downregulates ACE2 expression, increasing AngII, which activates nuclear factor-κB (NF-κB) and promotes TNF-α production as well as shedding of the soluble IL-6Ra via disintegrin and metalloprotease 17 (ADAM17). Ultimately, dysregulated host responses to SARS-CoV-2 drive excessive inflammation and severe disease progression [18]. Vaccination may attenuate cytokine storm responses, thereby potentially decreasing the risk of thrombosis and mortality.

It was well established that older individuals with SARS-CoV-2 experience more severe disease and higher mortality than younger people [19]. Older age was also associated with greater risk of cardiovascular complications and death. In our cohort, the mean age was 66.5 years, consistent with previous research reporting ages of 60–75 years [6,7,8,9,13,14,15,16,17]. Our study found that patients who received the COVID-19 vaccine had lower cumulative all-cause mortality rates over time compared to previously reported data. Specifically, the 1-year, 2-year, and 3-year mortality rates were 6.9%, 10.9%, and 15.3%, respectively. These figures are significantly lower than the previously reported 23–40% mortality rate associated with ALI following SARS-CoV-2 infection [6,7,8,9,13,14,15,16,17]. Our study aligns with prior research in the age distribution of patients but observes a notably lower mortality rate. Because PAOD predominantly affects older individuals, these findings may reflect the protective benefits associated with COVID-19 vaccination in this population. Specifically, vaccination was associated with a reduced mortality risk, underscoring its value as a preventative strategy for patients with PAOD. This reduction may be mediated by fewer major adverse cardiovascular events (MACEs) or acute respiratory distress syndrome (ARDS), both of which contribute to mortality in this group.

Across four hospitals in New York City, Bilaloglu et al. analyzed 3334 patients with COVID-19, among whom 365 had arterial thrombosis and 207 had venous thrombosis. The overall mortality rate was 24.5% [8]. In Spain, Gonzalez-Fajardo et al. retrospectively reviewed 2943 COVID-19 patients and identified 106 cases of ALI, with a reported mortality rate of 23.6% [16]. Bautista Sánchez et al. examined 30 patients with concurrent COVID-19 and ALI across six hospitals in Peru, reporting a 23.3% mortality rate and 30% amputation rate [13]. In Italy, Bellosta et al. reported 40% mortality among 20 COVID-19 patients with ALI [6]. In New York, Ilonzo et al. observed 33.3% mortality among 21 COVID-19 patients with ALI [17]. Goldman further noted that COVID-19 patients with worse respiratory symptoms accompanied by lower limb arterial thrombosis had a high amputation rate of 25% and a mortality rate of 38% [14]. In our cohort, vaccinated patients had cumulative lower limb amputation rates of 0.9% at 1 year, 1.5% at 2 years, and 1.9% at 3 years. These are notably lower than those reported in previous research. These findings suggest that COVID-19 vaccination may play a role in reducing lower limb amputation rates in patients with PAOD, potentially by attenuating prothrombotic and hyperinflammatory sequelae of infection and thereby protecting vital organs and extremities, and may mitigate the risk of limb-threatening events in patients with pre-existing PAOD. Table 3 summarizes the published studies on SARS-CoV-2 infection and acute limb ischemia, including mortality and amputation rates.

Klok et al. reported the deaths of 184 intensive care unit (ICU) patients with COVID-19 in the Netherlands, with pulmonary embolism (PE) being the most common thrombotic event [21]. Ackermann et al. reported widespread endothelial injury with inflammation, increased angiogenesis, and diffuse vascular thrombosis in COVID-19, indicating severe disruption of vascular integrity with microvascular clotting and alveolar capillary obstruction [22]. Previous research has shown that thromboembolic events and mortality correlated with the severity of SARS-CoV-2 infection [8]. Beyond arterial thrombosis, a systematic review of 86 studies involving 33,970 patients by Nopp et al. demonstrated an elevated risk of venous thromboembolism (VTE) in COVID-19. The overall VTE prevalence was 14.1%, with subgroup analysis showing 22.7% among ICU patients [23]. Coagulation abnormalities and coagulopathy likely contribute to VTE and deep vein thrombosis (DVT) [24]. A hypercoagulable state may impair kidney function, promote lower limb edema due to reduced venous return, and exacerbate ischemia. COVID-19 further complicates this condition by causing interstitial pneumonia [25], diminishing cardiopulmonary reserve, and promoting physical inactivity. This sedentary state increases the risk of DVT, further worsening lower limbs vascular complications. Severe respiratory involvement can progress to ARDS, which is a life-threatening complication and a major cause of death [6,13,17]

In our study, vaccinated individuals had a lower, non-significant lower risk of ischemic stroke (HR = 0.958; 95% CI = 0.902–1.019). SARS-CoV-2 enters cells via ACE2, which was highly expressed in human tissues such as the lungs, heart, kidneys, intestines, and vascular endothelium [18,26]. By contrast, ACE2 expression was relatively lower in the brain [26,27], which may explain the lower incidence of ischemic stroke observed in some COVID-19 cohorts. Using TriNetX data, Wang et al. reported that 4,131,717 COVID-19 survivors experienced a higher risk of stroke, including both hemorrhagic and ischemic types [28]. In contrast, Lu et al. conducted a study on 5,397,278 older US adults who received the COVID-19 bivalent vaccine and found no significant increase of stroke risk during the period immediately following vaccination [29].

Attisani et al. conducted a systematic review of 36 articles and indicated 194 patients with ALI, where the use of low-molecular-weight heparin (LMWH) appeared to have a dual role as an antimicrobial adjunct and in improving peripheral perfusion [9,17,30]. However, the review still revealed a high mortality rate of 33–40% [9,17,30]. Similarly, Ilonzo et al.’s systematic review of 60 studies demonstrated that patients with COVID-19 and ALI who were treated with anticoagulation strategies, including oral anticoagulants or LMWH, also exhibited a high mortality rate of 23–33% [13,16,17]. Furthermore, Bellosta et al. reported that prolonged systemic heparin might improve overall survival, but the mortality rate in that cohort was 40% [6]. Anticoagulant therapy carried bleeding complications, and individuals with higher bleeding risk were often also at greater risk for thrombotic events, as shown in studies of CAD populations [31]. These patients typically had multiple comorbidities that increase their risk of ischemia. When major bleeding occurred, interruption of antithrombotic therapy was often unavoidable and may subsequently elevated the risk of thrombosis. Beyond medication cessation, blood transfusions administered after major bleeding could provoke inflammatory responses that activate the body’s coagulation system. This activation increased the risk of developing thrombotic events [18]. When human blood is transfused, an increase in red blood cell (RBC) count leads to platelet recruitment and margination, bringing platelets closer to the thrombus and enhancing platelet deposition. Additionally, elevated hematocrit levels increased the frequency of platelet–thrombus interactions, accelerating the accumulation of platelets in the thrombus [32]. Both processes may contribute to higher mortality. In addition to anticoagulation strategies, preventing cytokine storm-driven thrombus formation through vaccination may be of greater importance.

### Strengths and Limitations

The strength of our study is that it is the first large-scale study to evaluate the impact of the COVID-19 vaccine on reducing complications in PAOD patients, including amputation rate and mortality. Given that SARS-CoV-2 infection is associated with a hypercoagulable state leading to arterial or venous thrombosis, the COVID-19 vaccine provides indirect protection and preventive effects for PAOD patients.

Our study had several limitations: First, during the COVID-19 pandemic, patients were often reluctant to visit hospitals due to fear of being diagnosed with COVID-19 and having to undergo quarantine policies. In the US, the vast geographic distribution of healthcare facilities presents challenges for older adults, particularly those with PAOD. Limited mobility and pain caused by PAOD make it difficult for these patients to seek timely medical care, leading to delays in diagnosis and treatment. This was especially true for PAOD patients, who were often mobility impaired. Consequently, some patients may have died at home without seeking medical care, potentially leading to an underestimation of the study numbers. Second, critically ill patients who were sedated and intubated in the ICU could not express issues related to lower limb pain or PAOD symptoms, which may lead to an underestimation of the study numbers. Third, there was potential for misclassification bias and residual confounding factors due to the use of electronic health records in the study. Fourth, we were unable to evaluate the number and type of vaccine doses administered using the US global network TriNetX database. Fifth, we did not know whether the symptomatic PAOD patient was treated with thrombectomy, stent placement, or bypass or if only low-molecular-weight heparin was used. Finally, the TriNetX database provided information only on all-cause mortality. It did not specify whether deaths following SARS-CoV-2 infection were due to MACEs or ARDS, leaving the exact cause of death unclear.

## 5. Conclusions

In summary, unvaccinated patients with PAOD had a 1.2-fold higher mortality and a 1.4-fold higher lower limb amputation than vaccinated patients. This protective effect may be linked to attenuation of cytokine storm responses and modulation of prothrombotic pathways, helping to safeguard vital organs such as the heart, lungs, kidneys, and the extremities. We recommend promoting regular physical activity and discouraging sedentary behavior among PAOD patients. Additionally, encouraging COVID-19 vaccination in this population could help prevent further complications and reduce the risk of death.

## Figures and Tables

**Figure 1 vaccines-13-00969-f001:**
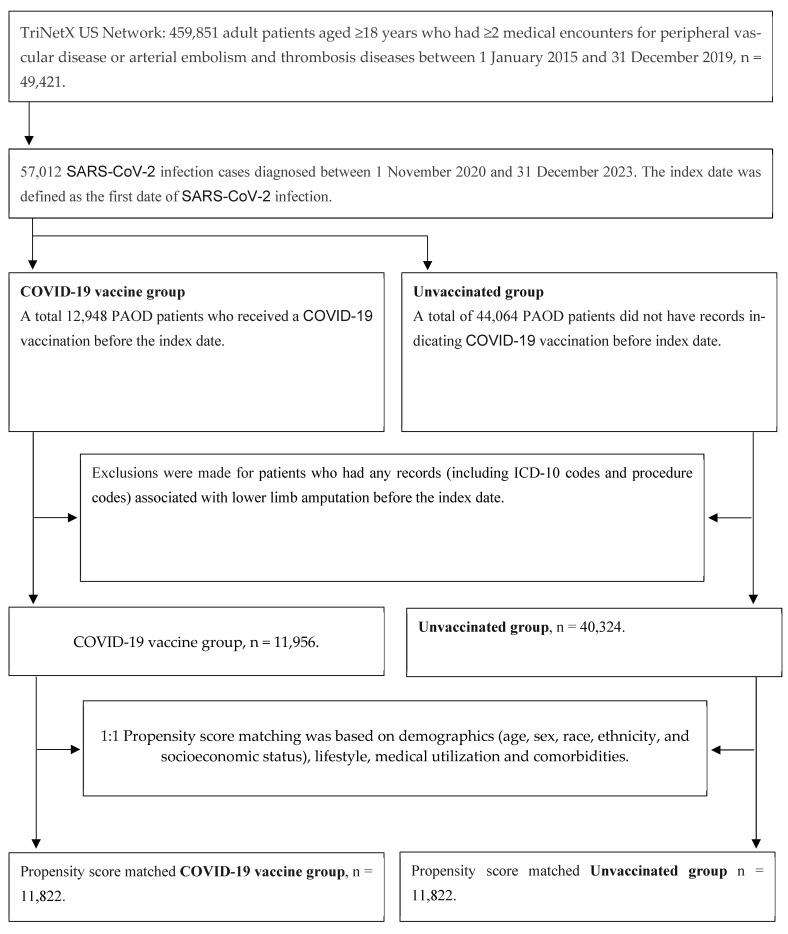
Study flowchart for sample selection.

**Figure 2 vaccines-13-00969-f002:**
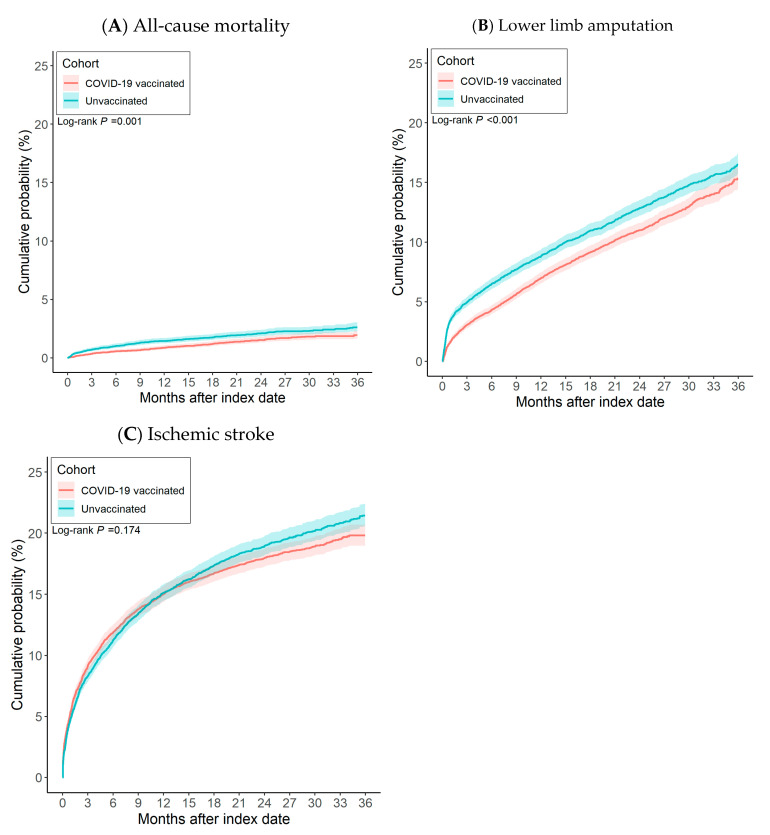
The risk of (**A**) all-cause mortality, (**B**) lower limb amputation, and (**C**) ischemic stroke between patients with COVID-19 vaccination and without COVID-19 vaccination.

**Table 1 vaccines-13-00969-t001:** Baseline characteristics.

	Before PSM			After PSM		
	With Vaccine	Without Vaccine	SMD	With Vaccine	Without Vaccine	SMD
N	11,956	40,324		11,822	11,822	
Age at index date	66.5 ± 14.1	65.7 ± 14.9	0.0554	66.5 ± 14.1	66.4 ± 14.3	0.0039
Sex						
Female	6652 (55.6%)	21,284 (52.8%)	0.0573	6569 (55.6%)	6608 (55.9%)	0.0066
Male	4900 (41.0%)	15,813 (39.2%)	0.0361	4849 (41.0%)	4813 (40.7%)	0.0062
Ethnicity						
Not Hispanic or Latino	10,448 (87.4%)	30,345 (75.3%)	0.3152	10,321 (87.3%)	10,338 (87.4%)	0.0043
Hispanic or Latino	832 (7.0%)	2368 (5.9%)	0.0443	825 (7.0%)	821 (6.9%)	0.0013
Race						
White	9360 (78.3%)	29,012 (71.9%)	0.1470	9252 (78.3%)	9261 (78.3%)	0.0018
Black or African American	1374 (11.5%)	4708 (11.7%)	0.0057	1370 (11.6%)	1372 (11.6%)	0.0005
Asian	299 (2.5%)	527 (1.3%)	0.0874	280 (2.4%)	291 (2.5%)	0.0061
Socioeconomic and psychosocial circumstances	2068 (17.3%)	4040 (10.0%)	0.2131	2005 (17.0%)	2012 (17.0%)	0.0016
BMI						
<30.0	5946 (49.7%)	16,350 (40.5%)	0.1854	5839 (49.4%)	5926 (50.1%)	0.0147
30.0–34.9	2891 (24.2%)	8244 (20.4%)	0.0898	2850 (24.1%)	2800 (23.7%)	0.0099
≥35.0	2536 (21.2%)	7276 (18.0%)	0.0798	2503 (21.2%)	2488 (21.0%)	0.0031
Lifestyle						
Nicotine dependence	3269 (27.3%)	5213 (12.9%)	0.3654	3139 (26.6%)	3115 (26.3%)	0.0046
Alcohol-related disorders	716 (6.0%)	1581 (3.9%)	0.0954	690 (5.8%)	689 (5.8%)	0.0004
Baseline medical utilization						
Preventive medicine services	1094 (9.2%)	2359 (5.9%)	0.1255	1055 (8.9%)	1022 (8.6%)	0.0099
Emergency department services	3666 (30.7%)	13,376 (33.2%)	0.0538	3644 (30.8%)	3597 (30.4%)	0.0086
Inpatient encounter	3425 (28.6%)	11,459 (28.4%)	0.0051	3402 (28.8%)	3353 (28.4%)	0.0092
Baseline comorbidity						
Hypertensive diseases	8672 (72.5%)	26,705 (66.2%)	0.1371	8555 (72.4%)	8576 (72.5%)	0.0040
Disorders of lipoprotein metabolism	7247 (60.6%)	21,973 (54.5%)	0.1241	7140 (60.4%)	7121 (60.2%)	0.0033
Diabetes mellitus	5143 (43.0%)	15,269 (37.9%)	0.1051	5065 (42.8%)	5015 (42.4%)	0.0086
T2DM with circulatory complications	1318 (11.0%)	4091 (10.1%)	0.0286	1302 (11.0%)	1255 (10.6%)	0.0128
T1DM with circulatory complications	31 (0.3%)	162 (0.4%)	0.0248	31 (0.3%)	33 (0.3%)	0.0033
Chronic lower respiratory diseases	4236 (35.4%)	12,529 (31.1%)	0.0926	4156 (35.2%)	4113 (34.8%)	0.0076
Ischemic heart diseases	4806 (40.2%)	14,830 (36.8%)	0.0703	4735 (40.1%)	4735 (40.1%)	0.0000
Heart failure	3108 (26.0%)	9321 (23.1%)	0.0670	3059 (25.9%)	2978 (25.2%)	0.0157
Cerebrovascular diseases	2712 (22.7%)	7611 (18.9%)	0.0940	2667 (22.6%)	2611 (22.1%)	0.0114
Chronic kidney disease (CKD)	3389 (28.3%)	10,433 (25.9%)	0.0556	3346 (28.3%)	3323 (28.1%)	0.0043
Diseases of liver	1775 (14.8%)	4037 (10.0%)	0.1469	1727 (14.6%)	1656 (14.0%)	0.0172
Osteoporosis without current pathological fracture	1449 (12.1%)	3238 (8.0%)	0.1362	1389 (11.7%)	1357 (11.5%)	0.0084
Osteoporosis with current pathological fracture	121 (1.0%)	271 (0.7%)	0.0372	114 (1.0%)	111 (0.9%)	0.0026
Rheumatoid arthritis with rheumatoid factor	258 (2.2%)	508 (1.3%)	0.0693	241 (2.0%)	242 (2.0%)	0.0006
Other rheumatoid arthritis	644 (5.4%)	1555 (3.9%)	0.0729	619 (5.2%)	614 (5.2%)	0.0019
Non-pressure chronic ulcer of lower limb	474 (4.0%)	1595 (4.0%)	0.0005	470 (4.0%)	461 (3.9%)	0.0039
Gangrene	191 (1.6%)	361 (0.9%)	0.0633	174 (1.5%)	174 (1.5%)	0.0000

Baseline period was defined as the time interval within 6 months before index date. Propensity score matching was based on demographics (age, sex, race, ethnicity, and socioeconomic status), lifestyle, medical utilization, and comorbidities. Data are presented as the mean ± SD for continuous variables (e.g., age) and n (%) for categorical variables. SMD (standardized mean difference): measure of between-group imbalance independent of sample size; |SMD| < 0.10 indicates negligible imbalance.

**Table 2 vaccines-13-00969-t002:** The cumulative probability and hazard ratio for risk of study event after index date among propensity score matched patients receiving COVID-19 vaccine and unvaccinated individuals.

		Cumulative Probability (95% CI) of Study Outcomes			
	Event	1-Year	2-Year	3-Year	Hazard Ratio (95% CI)
Risk of all-cause mortality					
Vaccinated (*n* = 11,822)	1363	6.9% (6.5–7.4%)	10.9% (10.4–11.5%)	15.3% (14.4–16.2%)	0.857 (0.796–0.922)
Unvaccinated (*n* = 11,822)	1494	8.8% (8.2–9.3%)	12.7% (12.1–13.4%)	16.3% (15.5–17.1%)	Reference
Risk of lower limb amputation					
Vaccinated (*n* = 11,822)	174	0.9% (0.7–1.1%)	1.5% (1.3–1.8%)	1.9% (1.6–2.2%)	0.716 (0.587–0.873)
Unvaccinated (*n* = 11,822)	226	1.5% (1.2–1.7%)	2.1% (1.8–2.4%)	2.6% (2.3–3.0%)	Reference
Risk of ischemic stroke					
Vaccinated (*n* = 11,822)	2056	15.0% (14.3–15.6%)	17.9% (17.2–18.6%)	19.8% (19.0–20.7%)	0.958 (0.902–1.019)
Unvaccinated (*n* = 11,822)	2041	15.1% (14.4–15.8%)	18.9% (18.1–19.6%)	21.4% (20.5–22.3%)	Reference

CI, confidence interval.

**Table 3 vaccines-13-00969-t003:** Acute limb ischemia mortality and amputation rates after SARS-CoV-2 infection.

Author	COVID-19 Patient	ALI Patient	Age	Sex (Male%)	Study Population	Mortality Rate (%)	Amputation Rate (%)
Juan Bautista S_anchez et al., 2021 [13]		30	60	76.6%	Six hospitals in Peruvian	23.3%	30%
Bellosta et al., 2020 [6]		20	75	90%	Italian	40%	
Seda Bilaloglu et al., 2020 [8]	3334	365	64	60.4%	4 hospitals in New York City	24.5%	
Goldman et al., 2020 [14]	16	15	70	56%	Montefiore Medical Center	38%	25%
Estefania Cantador et al., 2020 [15]	14	3	73	78.6%	Spain	28.6%	
Jose A. Gonzalez-Fajardoa et al., 2020 [16]	2943	106	65	67.92%	Spain	23.6%	
Ilonzo et al., 2021 [17]		21	64.6	52.4%	New York	33.3%	
Jeffrey E. Indes et al., 2021 [9]	424	15	64	66.7%	New York City	40%	
Yana Etkin., et al., 2021 [20]	12,630	49	67	76%	United States	43%	10%

ALI: acute limb ischemia.

## Data Availability

The data that support the findings of this study are available from the TriNetX Analytics Network, available online: https://trinetx.com (accessed on 17 July 2024).

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
