# Peer review of "COVID-19 Vaccination Reduces Lower Limb Amputation Rates and Mortality Rate in Patients with Pre-Existing Peripheral Vascular Disease Based on TriNetX Database"

_vaccines, 2025, doi:10.3390/vaccines13090969_

Round 1

Reviewer 1 Report

Comments and Suggestions for Authors

The aim of this study was to determine the impact of COVID-19 vaccination on reducing complications such as lower limb amputation, ischemic stroke, and mortality in patients with preexisting peripheral vascular disease (PAOD) infected with SARS-CoV-2. This study assessed the potential benefits of COVID-19 vaccination in improving the prognosis of patients with PAOD. Data for the study were obtained on July 17, 2024, from the TriNetX network in the USA.

Due to the established association between COVID-19 and thromboembolic complications, this topic has high practical value.

The positive aspects of this study include: matching patients to propensity scores for covariates, a large number of patients (>23,000) after matching, precise description of the cohorts, and good group distribution after matching.

Notes:

• In the title and Table 3, "moRtality" is used instead of "motality."

• There is no information on the number of vaccine doses (e.g., incomplete vaccination, primary vaccination, or booster vaccination), which reduces the accuracy of the conclusions.

• The vaccine type (mRNA vs. vector vaccine) was not included. • The direct causality of deaths (e.g., MACE vs. ARDS) was not assessed.

• References should be provided in parentheses, not superscripts.

2.1. Data Source: The notation is ambiguous and confusing, as the reader perceives a contradiction. - […] it was exempt from Institutional Review Board (IRB) approval. - […] was approved by the Institutional Review Board of Chung Shan Medical University Hospital (CSMUH), under approval number CS2-21176.

2.2. Study Population: The notation is inconsistent, e.g., "ICD-10: I73.0" and "ICD-10: I74."

Please standardize the notation of the numbers.

2.5. Outcomes: The wording is unclear, please correct it: "For each participant, follow-up started at the index date (defined as the date of COVID-19 diagnosis) and ended at the earliest occurrence of the study outcomes or censoring at the last recorded fact within the 3-year time window in the patient's record."

Table 1 – missing explanation: SMD, Age at index date - arithmetic mean, median?

Figure 2 – please increase the font size to make it legible, especially "p". Months would be more informative on the X-axis.

3. Results: "The hazard ratios were 0.857 (95% confidence interval [CI], 0.796–0.922) [...]" Please standardize the notation in parentheses to "95% CI, 0.796–0.922".

Why was HR calculated and not the Odds ratio?

4. Discussion: It is worth expanding the discussion to include possible biological mechanisms of protection: antithrombotic effects, reduction of cytokine storm, endothelial stabilization.

6. Conclusion: "In summary, our study demonstrates that patients with PAOD who received the COVID-19 vaccine had lower rates of both amputation and mortality."

In your conclusions, please clarify the extent to which these phenomena were observed, e.g., was the difference two-fold or ten-fold?

Author Response

Dear Reviewer: 1

We sincerely thank the reviewer 1. for insightful suggestions, which have truly enhanced the clarity and impact of our manuscript.

Comment 1: In the title and Table 3, "moRtality" is used instead of "motality."

Response 1: After revision: we change to motality as your excellent suggestion. Therefore, I have revised manuscript on (page 1, Line 1) and (page 23, Line 1).

Comment 2: There is no information on the number of vaccine doses (e.g., incomplete vaccination, primary vaccination, or booster vaccination), which reduces the accuracy of the conclusions.

Comment 3: The vaccine type (mRNA vs. vector vaccine) was not included.

Comment 4: The direct causality of deaths (e.g., MACE vs. ARDS) was not assessed.

Response 2 and 3: We thank the reviewer for these insightful comments. We argree that providing detailed information on vaccine dosage and type is crucial for interpreting the study's conclusions. In response to these valuable suggestions, we have performed the following sensitivity analyses:

  • Analysis by Vaccine Dose: We have stratified the analysis by the number of vaccine doses received prior to the index date, comparing the unvaccinated group with those who received 1, 2, or ≥3 (booster) doses.
  • Analysis by Vaccine Type: We have further stratified the analysis by vaccine type (BNT162b2, mRNA-1273, and adenoviral vector). This includes comparisons of each vaccine type against the unvaccinated cohort, as well as head-to-head comparisons between different vaccine types (e.g., BNT162b2 vs. mRNA-1273).

These new analyses were conducted using propensity score matching within each stratum to ensure robust comparisons.

We have now included a detailed description of these methods in the Methods section under " Statistical Analysis ".(Page 7, Lines 37-38, Page 8, Lines 1-6) and have presented the corresponding findings in the Results section (Page 8, Lines 23-38,  Page 9, Lines 1-19, and Supplementary Table S1-Supplementary Table S6)

However, on the TriNetX platform, vaccine dose information (including incomplete vaccination, primary vaccination, and booster doses) cannot be accuracy captured. In the United States, many individuals receive vaccines at local pharmacies rather than hospitals, and such administrations are often not documented in medical records. We have therefore acknowledged this as a limitation of our study. In the paragraph of Strengths and Limitations, we add “Our study had several limitations: ……Fourth, we were unable to evaluate the number and type of vaccine doses administered using the US network TriNetX database.”……(page 13, lines 16-18)

Response 4: On the TriNetX platform, we were only able to obtain data on all-cause motality, and cause-specific motality rates (e.g., MACE vs. ARDS) were not available. In the paragraph of Strengths and Limitations, we mentioned “Our study had several limitations: …… Finally, the TriNetX database provided information only on all-cause motality. It did not specify whether deaths following SARS-CoV-2 infection were due to MACE or ARDS, leaving the exact cause of death unclear.”…… (page 13, lines 20-23)

Comment 5: References should be provided in parentheses, not superscripts.

Response 5: We have adopted the use of parentheses in the manuscript, following your excellent suggestion. (page 4, line 1 to page 13, line 33)

Comment 6: Data Source: The notation is ambiguous and confusing, as the reader perceives a contradiction. - […] it was exempt from Institutional Review Board (IRB) approval. - […] was approved by the Institutional Review Board of Chung Shan Medical University Hospital (CSMUH), under approval number CS2-21176.

Response 6: Regarding the Data Source, we accessed the TriNetX platform through Chung Shan Medical University, with approval from the Institutional Review Board of Chung Shan Medical University Hospital (CSMUH) , Taichung, Taiwan under approval number CS2-21176. In line with your excellent suggestion, we deleted the statement indicating that “it was exempt from Institutional Review Board (IRB) approval.” (page 5, lines 31-33)

Comment 7: Study Population: The notation is inconsistent, e.g., "ICD-10: I73.0" and "ICD-10: I74." Please standardize the notation of the numbers.

Response 7: Regarding the Study Population, we identified patients with a history of peripheral vascular disease (including vasospasm or arterial thrombus–related conditions) following SARS-CoV-2 vaccination. In total, 459,851 adult patients aged ≥18 years were included, each having at least two medical encounters for peripheral vascular disease (ICD-10: I73.0) or arterial embolism and thrombosis (ICD-10: I74) in relation to arterial events. (page 5, lines 36-38, page 6, line 1)

Comment 8: Outcomes: The wording is unclear, please correct it: "For each participant, follow-up started at the index date (defined as the date of COVID-19 diagnosis) and ended at the earliest occurrence of the study outcomes or censoring at the last recorded fact within the 3-year time window in the patient's record."

Response 8: Thank you for the suggestion. We revised the Outcomes paragraph to explicitly define the time origin and all censoring rules. The revised text now states that “For each participant, follow-up began on the index date (defined as the date of COVID-19 diagnosis) and ended at the earliest of the outcomes event or censoring at the last recorded observation within 3 years of the index date.” (page 6, lines 28-30 )

Comment 9: Table 1 – missing explanation: SMD, Age at index date - arithmetic mean, median?

Response 9: In Table 1, we now clarify that continuous variables (including “Age at index date”) are presented as arithmetic mean ± standard deviation (SD), whereas categorical variables are presented as n (%). We also added a footnote defining SMD (standardized mean difference) and the balance criterion (absolute SMD < 0.10 indicates negligible imbalance). (page 20, lines 21-22)

Comment 10: Figure 2 – please increase the font size to make it legible, especially "p". Months would be more informative on the X-axis.

Response 10: Figure 2- We already increase the font size to make it legible, especially "p". Chang to Months on the X-axis as your helpful suggestion. (page 22)

Comment 11: Results: "The hazard ratios were 0.857 (95% confidence interval [CI], 0.796–0.922) [...]" Please standardize the notation in parentheses to "95% CI, 0.796–0.922". Why was HR calculated and not the Odds ratio?

Response 11: Regarding the Results, we change to …”(95% CI, 0.796–0.922)”… as your excellent suggestion. Because this is a time-to-event analysis, we used hazard ratios (HRs) rather than odds ratios (ORs).”

Comment 12: Discussion: It is worth expanding the discussion to include possible biological mechanisms of protection: antithrombotic effects, reduction of cytokine storm, endothelial stabilization.

Response 12: Regarding the possible biological mechanisms of protection: antithrombotic effects, reduction of cytokine storm, endothelial stabilization, in the paragraph of Discussion, we add ……“SARS-CoV-2 enters host cells via binding of its spike proteins to the ACE2 receptor and proteolytic priming by the host protease TMPRSS2. Following entry into respiratory epithelial cells, SARS-CoV-2 activates the immune system, inducing a broad cytokine response while eliciting a comparatively weak interferon response. The immune activation stimulates proinflammatory Th1 cells and CD14+CD16+ monocytes, amplifying signaling pathways that recruit macrophages and neutrophils to the sites of infection. The resulting feed-forward cascade culminates in a cytokine storm, contributing to severe lung inflammation.(18) SARS-CoV-2 rapidly activates pathogenic Th1 cells to produce granulocyte macrophage colony‐stimulating factor (GM-CSF) and interleukin‐6 (IL-6), GM-CSF further stimulates CD14+CD16+ monocytes to release of tumor necrosis factor‐α(TNF-α) and IL-6. Membrane-bound immune receptors, including Fc and Toll-like receptors, contribute to this inflammation amplification, while weak interferon-γ (IFN-γ) responses permit continued cytokine production. Neutrophil extracellular traps also promote cytokine release. The cytokine storm, characterized by elevated IL-6 and TNF-α levels, is linked to dysregulated angiotensin 2 (AngII) signaling. SARS-CoV-2 downregulates ACE2 expression, increasing AngII, which activates nuclear factor‐κB (NF-κB) and promotes TNF-α production as well as shedding of the soluble IL‐6Ra via disintegrin and metalloprotease 17 (ADAM17). Ultimately, dysregulated host responses to SARS-CoV-2 drive excessive inflammation and severe disease progression.(18) Vaccination may attenuate cytokine storm responses, thereby potentially decreasing the risk of thrombosis and mortality.”…(page 9, lines 32-38, page 10, lines 1-15)

We also add ……“Attisani et al. conducted a systematic review of 36 articles and indicated 194 patients with ALI, the use of low molecular weight heparin (LMWH) appeared to have a dual role in antimicrobial adjunct and improving peripheral perfusion.(9.17.27) However, the review still revealed a high mortality rate of 33-40%.(9.17.27) Similarly, Ilonzo et al.'s systematic review of 60 studies demonstrated that patients with COVID-19 and ALI who were treated with anticoagulation strategies, including oral anticoagulants or LMWH, also exhibited a high mortality rate of 23%-33%.(13.16.17) Furthermore, Bellosta et al. reported that prolonged systemic heparin might improve overall survival, but the mortality rate in that cohort was 40%.(6) Anticoagulant therapy carried bleeding complications, and individuals at higher bleeding risk were often also at greater risk for thrombotic events, as shown in studies of CAD populations.(28) These patients typically had multiple comorbidities that increase their risk of ischemia. When major bleeding occurred, interruption of antithrombotic therapy was often unavoidable and may subsequently elevated the risk of thrombosis. Beyond medication cessation, blood transfusions administered after major bleeding could provoke inflammatory responses that activate the body's coagulation system. This activation increased the risk of developing thrombotic events .(18) When human blood is transfused, an increase in red blood cell (RBC) count lead to platelet recruitment and margination, bringing platelets closer to the thrombus and enhancing platelet deposition. Additionally, elevated hematocrit levels increased the frequency of platelet-thrombus interactions, accelerating the accumulation of platelets in the thrombus.(29) Both processes may contribute to higher mortality. In addition to anticoagulation strategies, preventing cytokine storm-driven thrombus formation through vaccination may be of greater importance.”…(page 12, lines 10-33)

Comment 13: Conclusion: "In summary, our study demonstrates that patients with PAOD who received the COVID-19 vaccine had lower rates of both amputation and mortality." In your conclusions, please clarify the extent to which these phenomena were observed, e.g., was the difference two-fold or ten-fold?

Response 13: Regarding Conclusion, we change to…… “In summary, unvaccinated patients with PAOD had a 1.2-fold higher mortality and a 1.4-fold higher lower limb amputation than vaccinated patients.”……(page 13, lines 26-27)

Reviewer 2 Report

Comments and Suggestions for Authors

The manuscript titled SARS-CoV-2 vaccination reduces lower limb amputation rates and motality rate in patients with preexisting peripheral vascular disease by TriNetX database” is a retrospective cohort study on patients with peripheral arterial occlusive disease (PAOD), the all-cause mortality, lower limb amputation and risk of ischemia after  COVID-19 vaccination was compared to unvaccinated individuals. The study clearly demonstrated a significant benefit related to all-course mortality and lower limb amputation. However, no significant differences were seen for ischemic stroke.

 The manuscript is generally well written and should be of interest for the wider scientific community and clinicians. However, before being considered for acceptance, the points listed below should be addressed.

General comments

The authors fail to mention what type of COVID-19 vaccine was used. As the types of COVID-19 vaccines include whole virus vaccines, viral vector based vaccines, mRNA-based vaccines, etc, it would be informative to briefly describe which approach was used. Moreover, nothing is mentioned of the number of doses administered.   

The statement “the first COVID-19 infection was reported in Wuhan, China, leading to the severe acute respiratory syndrome coronavirus 2 (SARS-CoV-2)” (P2, L13) is incorrect. The infection is caused by the severe acute respiratory syndrome coronavirus 2 (SARS-CoV-2) while the disease is COVID-19. Please revise! Moreover, the authors repeatedly use “COVID-19 infection” throughout the manuscript, which should be replaced by “SARS-CoV-2 infection”

The statement “ there were 40,324 patients with COVID-19 vaccination available for analysis” (P3, L34) does not make sense as to my understanding these individuals were not vaccinated!

The heading in Table 2 (P7) of “Cumulative probability (95% CI) of psychiatric disorders” does not make sense. Please, explain!

Table 3 (P9) needs revision. In some cases ALI patients are presented as numbers or %. Be consistent. The column “Sex” contains in some cases numbers and in other cases numbers and %. The column “Mortality” is difficult to read.

Specific comments

P1, L3: “Our assessed” > “We assessed”

P1, L7: Delete “were identified as the study population”

P1, L12: “ PAOD patients with and without COVID-19 vaccination” > “COVID-19 vaccinated PAOD patients and unvaccinated individuals”

P1, L21: “that Patients” > “that patients”

P2, L6: “1.2, lead to cardiovascular death.” > “ [1.2] leading to cardiovascular death.”

P2, L17: “Bellosta et al. was” > “Bellosta et al. were”

P2, L34: “ Recent study” > “A recent study”

P2, L64: “healthcare organizations (HCOs)” > “HCOs” (already defined on P2, L56)

P3, L13: Add “Taichung, Taiwan” after “(CSMUH),”

P3, L30: “Without COVID-19 vaccine group” > “Unvaccinated group”

P4, Table 1 heading: “baseline” > “Baseline”

P6, Fig. 1: “Without COVID-19 vaccine group” > “Unvaccinated group” (occurs 3 times)

P7, Fig. 2: “Without COVID-19 vaccine group” > “Unvaccinated group” (occurs 3 times)

P7, Fig. 2 heading: “the risk of (A) all-cause” > “The risk of (A) All-cause”

P7, Table 2 heading: “patients with COVID-19 vaccine and without COVID-19 vaccine” > “patients receiving COVID-19 vaccine and unvaccinated individuals”

P8, Table 2: Repeated “With vaccine cohort” and “Without vaccine cohort” should be revised to “Vaccinated” and “Unvaccinated”, respectively.

P8, Discussion, L2: “matched controls” > “matched unvaccinated controls”

P8, Last line: “reviewing 2,943 COVID-19 patients” > “on 2,943 COVID-19 patients”

P9, Table 3 heading: “after covid-19 infection” > “after SARS-CoV-2 infection”

P9, Table 3: “age” > “Age”; “sex” > “Sex”; “Motality” > “Mortalitiy”

P10, L1: “reported on184 ICU patients with COVID-19 in the Dutch, finding that 41 died,” > “reported the death of184 ICU patients with COVID-19 in the Netherlands,”

P10, L3: reported autopsy studies of COVID-19 patients have shown extensive damage to the blood vessel lining, known as endothelial injury, accompanied by inflammation.” > “reported that extensive damage to the blood vessel lining, known as endothelial injury accompanied by inflammation, was found in COVID-19 patients.”

P10, L5: “heightened” > “increased”

P10, L14: Delete “from Austria”

P10, L26: “lower the risk” “lower risk”

P10, L27: “Covid-19 infection” > “SARS-CoV-2 infection”

P10, L33: “ older adults” > “elderly adults”

P10, Strengths and Limitations, L6 and throughout the text: “US” and “United States” (L10) are used in parallel. Be consistent and use either or!

P10, Strengths and Limitations, L7: What does “Eastern countries” stand for?

P11, L2: “intensive care unit” has been previously abbreviated to “ICU” (P10, L1), where it should be defined.

P11, Conclusions, L3: “to reduce cytokine storm” > “to reduced cytokine storm”

Comments on the Quality of English Language

Can be improved as suggested above.

Author Response

We sincerely thank the reviewer 2. for insightful suggestions, which have truly enhanced the clarity and impact of our manuscript.

Comment 1: The authors fail to mention what type of COVID-19 vaccine was used. As the types of COVID-19 vaccines include whole virus vaccines, viral vector based vaccines, mRNA-based vaccines, etc, it would be informative to briefly describe which approach was used. Moreover, nothing is mentioned of the number of doses administered.   

Response 1: We thank the reviewer for these insightful comments. We argree that providing

detailed information on vaccine type is crucial for interpreting the study's conclusions. In response to these valuable suggestions, we have performed the following sensitivity analyses:

  • Analysis by Vaccine Type: We have further stratified the analysis by vaccine type (BNT162b2, mRNA-1273, and adenoviral vector). This includes comparisons of each vaccine type against the unvaccinated cohort, as well as head-to-head comparisons between different vaccine types (e.g., BNT162b2 vs. mRNA-1273).

These new analyses were conducted using propensity score matching within each stratum to ensure robust comparisons.

We have now included a detailed description of these methods in the Methods section under " Statistical Analysis ".(Page 7, Lines 37-38, Page 8, Lines 1-6) and have presented the corresponding findings in the Results section (Page 8, Lines 23-38,  Page 9, Lines 1-19, and Supplementary Table S1-Supplementary Table S6)

However, on the TriNetX platform, vaccine dose information (including incomplete vaccination, primary vaccination, and booster doses) cannot be accuracy captured. In the United States, many individuals receive vaccines at local pharmacies rather than hospitals, and such administrations are often not documented in medical records. We have therefore acknowledged this as a limitation of our study. In the paragraph of Strengths and Limitations, we add “Our study had several limitations: ……Fourth, we were unable to evaluate the number and type of vaccine doses administered using the US network TriNetX database.”…… (page 13, lines 16-18)

Comment 2: The statement “the first COVID-19 infection was reported in Wuhan, China, leading to the severe acute respiratory syndrome coronavirus 2 (SARS-CoV-2)” (P2, L13) is incorrect. The infection is caused by the severe acute respiratory syndrome coronavirus 2 (SARS-CoV-2) while the disease is COVID-19. Please revise! Moreover, the authors repeatedly use “COVID-19 infection” throughout the manuscript, which should be replaced by “SARS-CoV-2 infection”

Response 2: We have already replaced the terms with “SARS-CoV-2 infection” and “COVID-19 vaccine,” following your excellent suggestion. (page 4, lines 16-19)

Comment 3: The statement “ there were 40,324 patients with COVID-19 vaccination available for analysis” (P3, L34) does not make sense as to my understanding these individuals were not vaccinated!

Response 3: In the paragraph of Unvaccinated group(Without COVID-19 vaccine group), we change ……“, 40,324 unvaccinated patients were included in the analysis.” (page 6, lines 21-22)

Comment 4: The heading in Table 2 (P7) of “Cumulative probability (95% CI) of psychiatric disorders” does not make sense. Please, explain!

Response 4: We have changed “Cumulative probability (95% CI) of psychiatric disorders” to “Cumulative probability (95% CI) of study outcomes” from the heading in Table 2, as it was a typographical error. (page 21, line 3)

Comment 5: Table 3 (P9) needs revision. In some cases ALI patients are presented as numbers or %. Be consistent. The column “Sex” contains in some cases numbers and in other cases numbers and %. The column “Mortality” is difficult to read.

Response 5: We revised Table 3 (p. 9) so that ALI patients are presented as numbers, male sex is expressed as a percentage, and both mortality and amputation rates are also presented as percentages, in accordance with your helpful suggestion. (page 23)

Comment 6:

P1, L3: “Our assessed” > “We assessed”

P1, L7: Delete “were identified as the study population”

P1, L12: “ PAOD patients with and without COVID-19 vaccination” > “COVID-19 vaccinated PAOD patients and unvaccinated individuals”

P1, L21: “that Patients” > “that patients”

P2, L6: “1.2, lead to cardiovascular death.” > “ [1.2] leading to cardiovascular death.”

P2, L17: “Bellosta et al. was” > “Bellosta et al. were”

P2, L34: “ Recent study” > “A recent study”

P2, L64: “healthcare organizations (HCOs)” > “HCOs” (already defined on P2, L56)

P3, L13: Add “Taichung, Taiwan” after “(CSMUH),”

P3, L30: “Without COVID-19 vaccine group” > “Unvaccinated group”

P4, Table 1 heading: “baseline” > “Baseline”

P6, Fig. 1: “Without COVID-19 vaccine group” > “Unvaccinated group” (occurs 3 times)

P7, Fig. 2: “Without COVID-19 vaccine group” > “Unvaccinated group” (occurs 3 times)

P7, Fig. 2 heading: “the risk of (A) all-cause” > “The risk of (A) All-cause”

P7, Table 2 heading: “patients with COVID-19 vaccine and without COVID-19 vaccine” > “patients receiving COVID-19 vaccine and unvaccinated individuals”

P8, Table 2: Repeated “With vaccine cohort” and “Without vaccine cohort” should be revised to “Vaccinated” and “Unvaccinated”, respectively.

P8, Discussion, L2: “matched controls” > “matched unvaccinated controls”

P8, Last line: “reviewing 2,943 COVID-19 patients” > “on 2,943 COVID-19 patients”

P9, Table 3 heading: “after covid-19 infection” > “after SARS-CoV-2 infection”

P9, Table 3: “age” > “Age”; “sex” > “Sex”; “Motality” > “Mortalitiy”

P10, L1: “reported on184 ICU patients with COVID-19 in the Dutch, finding that 41 died,” > “reported the death of184 ICU patients with COVID-19 in the Netherlands,”

P10, L3: reported autopsy studies of COVID-19 patients have shown extensive damage to the blood vessel lining, known as endothelial injury, accompanied by inflammation.” > “reported that extensive damage to the blood vessel lining, known as endothelial injury accompanied by inflammation, was found in COVID-19 patients.”

P10, L5: “heightened” > “increased”

P10, L14: Delete “from Austria”

P10, L26: “lower the risk” “lower risk”

P10, L27: “Covid-19 infection” > “SARS-CoV-2 infection”

P10, L33: “ older adults” > “elderly adults”

P10, Strengths and Limitations, L6 and throughout the text: “US” and “United States” (L10) are used in parallel. Be consistent and use either or!

P10, Strengths and Limitations, L7: What does “Eastern countries” stand for?

P11, L2: “intensive care unit” has been previously abbreviated to “ICU” (P10, L1), where it should be defined.

P11, Conclusions, L3: “to reduce cytokine storm” > “to reduced cytokine storm”

Response 6: We have already revised this in the Specific Comments section, following your helpful suggestion.

Reviewer 3 Report

Comments and Suggestions for Authors

Recommendation: Major Revision

Potentially very useful, large-scale study evaluating the outcomes of COVID-19 vaccination in patients with PAOD, conducted with an appropriate data source and statistical analyses.

Nevertheless, the study and its conclusions would benefit greatly if the following information could be included in the analyses: Vaccine types used, number of doses, whether natural infection occurred, timing relative to infection, cause-specific mortality. Furthermore, in addition to comparisons with previous studies, the discussion should attempt to hypothesize the effects of vaccination on coagulation pathways, inflammation, and endothelial function.

Minor points:

Figure 2 and Table 3 should be corrected.

The manuscript needs language editing (including the title).

Author Response

Dear Reviewer: 3

We sincerely thank the reviewer 3. for insightful suggestions, which have truly enhanced the clarity and impact of our manuscript.

Comment 1: Potentially very useful, large-scale study evaluating the outcomes of COVID-19 vaccination in patients with PAOD, conducted with an appropriate data source and statistical analyses. Nevertheless, the study and its conclusions would benefit greatly if the following information could be included in the analyses: Vaccine types used, number of doses, whether natural infection occurred, timing relative to infection, cause-specific mortality. Furthermore, in addition to comparisons with previous studies, the discussion should attempt to hypothesize the effects of vaccination on coagulation pathways, inflammation, and endothelial function.

Response 1: We thank the reviewer for these insightful comments. We argree that providing

detailed information on vaccine dosage and type is crucial for interpreting the study's conclusions. In response to these valuable suggestions, we have performed the following sensitivity analyses:

  • Analysis by Vaccine Dose: We have stratified the analysis by the number of vaccine doses received prior to the index date, comparing the unvaccinated group with those who received 1, 2, or ≥3 (booster) doses.
  • Analysis by Vaccine Type: We have further stratified the analysis by vaccine type (BNT162b2, mRNA-1273, and adenoviral vector). This includes comparisons of each vaccine type against the unvaccinated cohort, as well as head-to-head comparisons between different vaccine types (e.g., BNT162b2 vs. mRNA-1273).

These new analyses were conducted using propensity score matching within each stratum to ensure robust comparisons.

We have now included a detailed description of these methods in the Methods section under " Statistical Analysis ".(Page 7, Lines 37-38, Page 8, Lines 1-6) and have presented the corresponding findings in the Results section (Page 8, Lines 23-38,  Page 9, Lines 1-19, and Supplementary Table S1-Supplementary Table S6)

However, on the TriNetX platform, vaccine dose information (including incomplete vaccination, primary vaccination, and booster doses) cannot be accuracy captured. In the United States, many individuals receive vaccines at local pharmacies rather than hospitals, and such administrations are often not documented in medical records. We have therefore acknowledged this as a limitation of our study. In the paragraph of Strengths and Limitations, we add “Our study had several limitations: ……Fourth, we were unable to evaluate the number and type of vaccine doses administered using the US network TriNetX database.”……

Response 1: Regarding the period of SARS-CoV-2 infection, In the paragraph of Study population, we mentioned ……“Initially, we identified 459,851 adult patients aged ≥18 years who had ≥2 medical encounters for peripheral vascular disease (ICD-10: I73.0, I73.1, I73.8, I73.9) or arterial embolism and thrombosis diseases (ICD-10: I74) between January 1, 2015, and December 31, 2019. From this population, we selected 57,012 SARS-CoV-2 infection cases (defined as ICD-10-CM U07.1 or PCR positive for SARS-CoV-2 RNA) that occurred between November 2020 and December 2023.”……(page 5, lines 36-38, page 6, lines 1-3)

Response 1: On the TriNetX platform, we were only able to obtain data on all-cause motality, and cause-specific motality rates (e.g., MACE vs. ARDS) were not available. In the paragraph of Strengths and Limitations, we mentioned “Our study had several limitations: …… Finally, the TriNetX database provided information only on all-cause motality. It did not specify whether deaths following SARS-CoV-2 infection were due to MACE or ARDS, leaving the exact cause of death unclear.”…… (page 13, lines 20-23)

Response 1: Regarding the hypothesize the effects of vaccination on coagulation pathways, inflammation, and endothelial function, in the paragraph of Discussion, we add ……“SARS-CoV-2 enters host cells via binding of its spike proteins to the ACE2 receptor and proteolytic priming by the host protease TMPRSS2. Following entry into respiratory epithelial cells, SARS-CoV-2 activates the immune system, inducing a broad cytokine response while eliciting a comparatively weak interferon response. The immune activation stimulates proinflammatory Th1 cells and CD14+CD16+ monocytes, amplifying signaling pathways that recruit macrophages and neutrophils to the sites of infection. The resulting feed-forward cascade culminates in a cytokine storm, contributing to severe lung inflammation.(18) SARS-CoV-2 rapidly activates pathogenic Th1 cells to produce granulocyte macrophage colony‐stimulating factor (GM-CSF) and interleukin‐6 (IL-6), GM-CSF further stimulates CD14+CD16+ monocytes to release of tumor necrosis factor‐α(TNF-α) and IL-6. Membrane-bound immune receptors, including Fc and Toll-like receptors, contribute to this inflammation amplification, while weak interferon-γ (IFN-γ) responses permit continued cytokine production. Neutrophil extracellular traps also promote cytokine release. The cytokine storm, characterized by elevated IL-6 and TNF-α levels, is linked to dysregulated angiotensin 2 (AngII) signaling. SARS-CoV-2 downregulates ACE2 expression, increasing AngII, which activates nuclear factor‐κB (NF-κB) and promotes TNF-α production as well as shedding of the soluble IL‐6Ra via disintegrin and metalloprotease 17 (ADAM17). Ultimately, dysregulated host responses to SARS-CoV-2 drive excessive inflammation and severe disease progression.(18) Vaccination may attenuate cytokine storm responses, thereby potentially decreasing the risk of thrombosis and mortality.”… (page 9, lines 32-38, page 10, lines 1-15)

We also add ……“Attisani et al. conducted a systematic review of 36 articles and indicated 194 patients with ALI, the use of low molecular weight heparin (LMWH) appeared to have a dual role in antimicrobial adjunct and improving peripheral perfusion.(9.17.27) However, the review still revealed a high mortality rate of 33-40%.(9.17.27) Similarly, Ilonzo et al.'s systematic review of 60 studies demonstrated that patients with COVID-19 and ALI who were treated with anticoagulation strategies, including oral anticoagulants or LMWH, also exhibited a high mortality rate of 23%-33%.(13.16.17) Furthermore, Bellosta et al. reported that prolonged systemic heparin might improve overall survival, but the mortality rate in that cohort was 40%.(6) Anticoagulant therapy carried bleeding complications, and individuals at higher bleeding risk were often also at greater risk for thrombotic events, as shown in studies of CAD populations.(28) These patients typically had multiple comorbidities that increase their risk of ischemia. When major bleeding occurred, interruption of antithrombotic therapy was often unavoidable and may subsequently elevated the risk of thrombosis. Beyond medication cessation, blood transfusions administered after major bleeding could provoke inflammatory responses that activate the body's coagulation system. This activation increased the risk of developing thrombotic events .(18) When human blood is transfused, an increase in red blood cell (RBC) count lead to platelet recruitment and margination, bringing platelets closer to the thrombus and enhancing platelet deposition. Additionally, elevated hematocrit levels increased the frequency of platelet-thrombus interactions, accelerating the accumulation of platelets in the thrombus.(29) Both processes may contribute to higher mortality. In addition to anticoagulation strategies, preventing cytokine storm-driven thrombus formation through vaccination may be of greater importance.”… (page 12, lines 10-33)

Comment 2: Figure 2 and Table 3 should be corrected.

Response 2: We have revised Figure 2 and Table 3 in accordance with your helpful suggestion.(page 22-24)

Comment 3: The manuscript needs language editing (including the title).

Response 3:We have revised manuscript as your helpful suggestion.

Reviewer 4 Report

Comments and Suggestions for Authors

The aim of the present study was to determine if prior COVID-19 vaccination confers protection against severe outcomes, specifically lower limb amputation, all-cause mortality, and ischemic stroke, in patients with pre-existing peripheral arterial occlusive disease (PAOD) who experience a SARS-CoV-2 infection by analyzing data from 48 healthcare organizations within the TriNetX Research Network's US Collaborative Network. This study underscores the critical importance of proactive vaccination in patients with PAOD and other vascular conditions to mitigate life- and limb-threatening risks. This work paves the way for future research into vaccines' effects on a broader range of cardiovascular and thrombotic complications, potentially mediated through the reduction of systemic inflammation and hypercoagulability.

Overall I found the manuscript well written and informative. My comments to improve manuscript:

  1. The authors need to address the potential impact of different vaccine types/doses (mRNA vs adenoviral, booster effects). Type of vaccine (mRNA vs adenoviral), number of doses, booster status, and time since vaccination were not available, which may influence outcomes.
  2. Limitation: The study relies on electronic health records, which may contain missing data, miscoding, or inconsistent definitions.
  3. Disease-specific severity adjustment – to enhance the novelty of the study it would be good to stratify results by PAOD severity (e.g., Fontaine or Rutherford stage) to see if benefits vary with disease stage.

Overall, I approve publication once the above comments are addressed.

Author Response

Dear Reviewer: 4

We sincerely thank the reviewer 4. for insightful suggestions, which have truly enhanced the clarity and impact of our manuscript.

Comment 1: The authors need to address the potential impact of different vaccine types/doses (mRNA vs adenoviral, booster effects). Type of vaccine (mRNA vs adenoviral), number of doses, booster status, and time since vaccination were not available, which may influence outcomes.

Response 1: We thank the reviewer for these insightful comments. We argree that providing detailed information on vaccine dosage and type is crucial for interpreting the study's conclusions. In response to these valuable suggestions, we have performed the following sensitivity analyses:

  • Analysis by Vaccine Dose: We have stratified the analysis by the number of vaccine doses received prior to the index date, comparing the unvaccinated group with those who received 1, 2, or ≥3 (booster) doses.
  • Analysis by Vaccine Type: We have further stratified the analysis by vaccine type (BNT162b2, mRNA-1273, and adenoviral vector). This includes comparisons of each vaccine type against the unvaccinated cohort, as well as head-to-head comparisons between different vaccine types (e.g., BNT162b2 vs. mRNA-1273).

These new analyses were conducted using propensity score matching within each stratum to ensure robust comparisons.

We have now included a detailed description of these methods in the Methods section under " Statistical Analysis ".(Page 7, Lines 37-38, Page 8, Lines 1-6) and have presented the corresponding findings in the Results section (Page 8, Lines 23-38,  Page 9, Lines 1-19, and Supplementary Table S1-Supplementary Table S6)

However, on the TriNetX platform, vaccine dose information (including incomplete vaccination, primary vaccination, and booster doses) cannot be accuracy captured. In the United States, many individuals receive vaccines at local pharmacies rather than hospitals, and such administrations are often not documented in medical records. We have therefore acknowledged this as a limitation of our study. In the paragraph of Strengths and Limitations, we add “Our study had several limitations: ……Fourth, we were unable to evaluate the number and type of vaccine doses administered using the US network TriNetX database.”…… (page 13, lines 16-18)

Comment 2: Limitation: The study relies on electronic health records, which may contain missing data, miscoding, or inconsistent definitions.

Response 2: This was a retrospective cohort study using the TriNetX platform, which relies on electronic health records and may be subject to missing data and miscoding. To address these limitations, future randomized controlled trials or animal studies may be designed to further investigate the effects of different vaccine types on the risk of thromboembolic complications. In the paragraph of Strengths and Limitations, we mentioned: Our study had several limitations:….. Third, there was potential for misclassification bias and residual confounding factors due to the use of electronic health records in the study….(page 13, lines 15-16)

Comment 3: Disease-specific severity adjustment – to enhance the novelty of the study it would be good to stratify results by PAOD severity (e.g., Fontaine or Rutherford stage) to see if benefits vary with disease stage.

Response 3: In this retrospective cohort study using the TriNetX platform, it was difficult to determine vaccination status across different severities of PAOD (e.g., Fontaine or Rutherford stages). We were only able to identify previous peripheral vascular disease based on ICD-10 codes (I73.0, I73.1, I73.8, I73.9). A real-world medical chart review may therefore provide a more accurate evaluation of the potential benefits across varying disease stages.

Round 2

Reviewer 3 Report

Comments and Suggestions for Authors

The authors addressed all my concerns 

Reviewer 4 Report

Comments and Suggestions for Authors

The authors addressed my comments and issues raised. I approve publication.